# Effect of a New Tele-Rehabilitation Program versus Standard Rehabilitation in Patients with Chronic Obstructive Pulmonary Disease

**DOI:** 10.3390/jcm11010011

**Published:** 2021-12-21

**Authors:** Jose Cerdán-de-las-Heras, Fernanda Balbino, Anders Løkke, Daniel Catalán-Matamoros, Ole Hilberg, Elisabeth Bendstrup

**Affiliations:** 1Department of Respiratory Diseases and Allergy, Center for Rare Lung Diseases, Aarhus University Hospital, 8200 Aarhus, Denmark; karbends@rm.dk; 2Department of Research and Development, Physio R&D ApS, 2000 Frederiksberg, Denmark; fernanda@physiord.com; 3Department of Respiratory Medicine, Vejle Hospital, 7100 Vejle, Denmark; aloekke@gmail.com (A.L.); ole.hilberg@rsyd.dk (O.H.); 4Department of Communication and Media Studies, UC3M Medialab, Madrid University Carlos III, 28903 Madrid, Spain; dacatala@hum.uc3m.es; 5Health Research Centre, University of Almeria, 04120 Almeria, Spain

**Keywords:** tele-rehabilitation, COPD, virtual agent

## Abstract

In chronic obstructive pulmonary disease (COPD), rehabilitation is recommended, but attendance rates are low. Tele-rehabilitation may be key. We evaluate the effect of a tele-rehabilitation program vs. standard rehabilitation on COPD. A randomized, non-inferiority study comparing eight weeks of tele-rehabilitation (physiotherapist video/chat-consultations and workout sessions with a virtual-autonomous-physiotherapist-agent (VAPA)) and standard rehabilitation in stable patients with COPD. At baseline, after 8 weeks and 3 and 6 months of follow-up, 6 min walk test distance (6MWTD), 7-day pedometry, quality of life, exercise tolerance, adherence, patient satisfaction and safety were assessed. Fifty-four patients (70 ± 9 years, male 57%, FEV1% 34.53 ± 11.67, FVC% 68.8 ± 18.81, 6MWT 376.23 ± 92.02) were included. Twenty-seven patients were randomized to tele-rehabilitation. Non-inferiority in Δ6MWTD at 8 weeks (47.4 ± 31.4), and at 3 (56.0 ± 38.0) and 6 (95.2 ± 47.1) months follow-up, was observed. No significant difference was observed in 7-day pedometry or quality of life. In the intervention group, 6MWTD increased by 25% and 66% at 3 and 6 months, respectively; adherence was 81%; and patient satisfaction was 4.27 ± 0.77 (Likert scale 0–5). Non-inferiority between groups and high adherence, patient satisfaction and safety in the intervention group were found after rehabilitation and at 3 and 6 months of follow-up. Tele-rehabilitation with VAPA seems to be a promising alternative.

## 1. Introduction

Rehabilitation forms part of the standard treatment in COPD as it improves the quality of life and exercise capacity and reduces symptoms and mortality [1], as well as emergency visits and hospital admissions [2]. Drop-out rates of pulmonary rehabilitation (PR) attendance are high, up to 59% due to limited motivation (49.0%), transportation troubles (23.8%), COPD exacerbation (18.4%), job-related reasons (4.8%) and hospital admissions (4.1%) [1,3,4]. The SARS-CoV pandemic has pushed the health sector to a point where new digital procedures to interact with patients need to be rapidly explored, developed and implemented to safely support patient needs [5]. Tele-rehabilitation could be one of the tools to facilitate training for patients with chronic diseases by providing rehabilitation while maintaining physical distance. In a recent review, the usage of tele-rehabilitation in COPD demonstrates potential effectiveness, high patient acceptance and strong motivation to engage patients in physical activity [6]. Tele-rehabilitation is a method that is used today to treat, test and follow patients from a distance in order to empower them to cope with their short- and long-term impairments and help them to be physically, mentally, emotionally, vocationally and socially independent, thus improving and maintaining their quality of life. In another review, five different methods of tele-rehabilitation in pulmonary diseases are described: 1—videoconferencing; 2—telephone; 3—using a website with telephone support; 4—using a mobile application; 5—text message support with a mobile app [7]. Tele-rehabilitation was born with the first aim to reduce hospitalization time and to treat patients in rural areas [8], but now, in the COVID-19 pandemic times, it is introduced as an alternative to conventional rehabilitation and as an action to prevent infections and support the continuity of rehabilitation [9,10,11]. However, tele-rehabilitation in patients with chronic lung diseases and other diseases has not always shown improvements in usability, cost-effectiveness, adherence, safety, and patient and therapist satisfaction, and further research and development to deduct the value of tele-rehabilitation is needed. Focus group interviews have suggested combining face-to-face consultations with user-friendly platforms [12,13]. Previously, tele-rehabilitation has been shown to be feasible and positively accepted by patients and to improve functional capacity, breathlessness, quality of life and physical activity [14]. However, patients perceived the technology used as difficult [15,16,17,18,19]. The interaction between patients with COPD at home and the healthcare professionals at the hospital through tele-rehabilitation has developed as a dialogue forming the basis for collective learning processes and new relationships [20]. Previous studies on tele-rehabilitation initiatives in Scotland have shown tele-rehabilitation to be more cost-effective for patients living in distant areas compared to outreach or centralized models [21]. However, there is still a need of studies on new technologies and on the efficacy, cost-effectiveness and long-term benefits of tele-rehabilitation. The aim of this randomized, non-inferiority study was to compare a tele-rehabilitation platform based on virtual agent technology with standard rehabilitation in patients with COPD. The focus of this study is to use tele-rehabilitation as a tool to remotely treat COPD patients.

## 2. Methods and Materials

### 2.1. Design of the Research

The study was performed as a prospective, single-center, non-inferiority randomized, prospective clinical study comparing standard rehabilitation in patients with COPD with tele-rehabilitation. Randomization was performed digitally at www.randomization.com (accessed on 19 January 2017) [22] with subjects randomized into one block (reproducible using seed 9194). The protocol for the study was approved by the Danish Data Protection Agency (reference 2012-58-006) and the Central Denmark Region Committee on Health Research Ethics (reference 1-16-02-417-16). The trial was registered at clinicaltrial.gov (ID NCT03569384). The different objective tests were executed by an independent research nurse not involved in the study.

### 2.2. Participants

Patients with spirometry and physician-verified COPD from the outpatient clinic of the Department of Respiratory Diseases and Allergy, Aarhus University Hospital, Denmark, were randomized if they: (1) were >18 years of age, (2) had been referred for standard rehabilitation and (3) signed an informed consent form. Musculoskeletal abnormalities, dizziness, substantial sensory or motor impairments, dementia and/or severe comorbidities that impeded training were all exclusion factors (Appendix A). Oxygen therapy was not an exclusion criterion.

### 2.3. Tele-Rehabilitation

Patients were randomized 1:1 to 8 weeks of standard or tele-rehabilitation [23]. During follow-up, patients in the intervention group were provided with the opportunity to use a Virtual Autonomous Physiotherapist Agent (VAPA) on a daily basis without the supervision of a physiotherapist. At the start of the study, upon the completion of rehabilitation, and after three and six months of follow-up, relevant clinical parameters were collected. Tele-rehabilitation was delivered with VAPA, a Eurostars-funded platform built by a European collaboration (Physio R&D and Cortrium, Copenhagen, Denmark, and bookBeo, Le Faou, France and Laster Technologies, Paris, France) and two university hospitals (Aarhus University Hospital, Denmark, and Oulu University Hospital, Finland) [24]. The tele-rehabilitation program was originally developed based on feedback from patients with chronic cardiopulmonary diseases [12,13]. VAPA is both (a) a multidisciplinary software that allows therapists to design tailored tele-rehabilitation programs for patients by combining video consultations, e-learning packages, physical training regimens, online questionnaires, patient digital files and direct chat with patients all in one application [25] and (b) a mobile app that can be installed in smartphones or tablets connected directly to a biometric sensor attachable to the chest, arm or finger of the patient to collect data and adjust the rehabilitation program in real time (Figure 1) [26]. Table 1 summarizes the content of the tele-rehabilitation program.

### 2.4. Standard Rehabilitation

In the standard rehabilitation group, patients with COPD underwent a conventional standardized rehabilitation program, as implemented at Aarhus University Hospital. Patients attended 2 weekly group training sessions of 1 h at the hospital with instruction by a physiotherapist and 6 h of education about COPD and its treatment for 8 weeks [36].

### 2.5. Endpoints

The primary endpoint was the difference between groups in the 6 min walk test distance (6MWTD) [37] from baseline to the completion of rehabilitation. The secondary endpoints were differences between groups from baseline to follow-up three and six months after the end of rehabilitation in the following: 6MWTD, ActiGraph Monitor wGT3X-BT tracked pedometry for 7 days, counting the number of steps taken and total vector magnitude counts per minute (VMCPM) [38]. Patient-reported outcomes were recorded by the St. George’s Respiratory Questionnaire, which was used to measure the quality of life (SGRQ) [39], Instrumental Activity of Daily Living (IADL) [40] and the General Anxiety Disorder Score (GAD7) [41]. A pulmonary function test collected information about the forced expiratory volume in the first second (FEV1) and forced vital capacity (FVC). Patients randomly assigned to VAPA tele-rehabilitation rated their satisfaction with the treatment by answering a question on a 5-point Likert scale (1: “very unsatisfied” to 5: “very satisfied”) every time they trained. To calculate adherence, the training time per exercise set performed and the weekly average training time were recorded. This was compared to the shortest training time per week shown to be satisfactory for COPD patients. A minimum training duration of 60 min per week was set as the goal [42].

### 2.6. Statistics

The primary endpoint is based on a minimally important clinical difference of 35 m in the 6 MWT distance [43]. A 30% drop-out rate was anticipated [44]. Thus, a total of 54 patients (1:1 ratio) were recruited; 27 patients were enrolled in each group. This sample size allows the estimation of a 95% confidence interval bound to perform the non-inferiority test with 80% test power and 0.4 effect size. We decided to use a two-sided independent *t*-test to analyze the secondary endpoints because 1. It is a robust test even with no normal distribution; 2. We did not find any outliers; 3. There was no previous hypothesis on the direction of the effects for the secondary endpoints; and 4. We expected a substantial number of dropouts potentially preventing the use of an analysis of variance (ANOVA). R and IBM SPSS Statistics version 25 were used to conduct the statistical analysis. According to their randomized treatment group, all patients’ results were analyzed, regardless of their adherence to or the received intended treatment, characterizing the use of the intention-to-treat approach [45].

## 3. Results

### 3.1. Patients

A total of 95 COPD patients were screened, with 54 of them being enrolled in the study. The baseline demographics of eligible patients who declined to participate were similar to those who were included. Fifty-four patients with COPD were included from March 2017 to March 2019; 27 patients were randomized to standard rehabilitation and 27 to tele-rehabilitation with VAPA (Figure 2). Reasons for dropout are expressed in Appendix A. Gender distribution was equal in the two groups, but in the tele-rehabilitation group, patients were younger; more were ever-smokers; and they had, to some extent, a better quality of life (Table 2).

### 3.2. 6 Min Walk Test (6MWT)

In the 6MWT, patients receiving the normal hospital rehabilitation and those who received tele-rehabilitation with VAPA walked 387 and 434 m after 8 weeks of training, respectively, corresponding to a statistically non-significant difference of 47 m in favor of tele-rehabilitation with VAPA (*p* = 0.14). Mean differences between groups and confidence intervals were calculated (Appendix A), and the 35 m non-inferiority margin for the 6MWT was not crossed between baseline and after 8 weeks of rehabilitation and after 3 and 6 months of follow-up (Figure 3).

### 3.3. Pedometry and Quality of Life

No difference between groups with respect to SGRQ, IADL, GAD7 and 7-day pedometry was found (Appendix A).

### 3.4. Continued Use of VAPA in the TR with VAPA

Among patients who were randomized to VAPA tele-rehabilitation, ten patients decided to continue training with only the “workout sessions with VAPA” after the first 8 weeks training, and five of them decided to continue training after the 3-month follow-up.

### 3.5. Exercise Set Time, Adherence, Patient Satisfaction and Safety

The exercise set mean time showed a difference of a 4 min (+25%) positive increment from 8 weeks (*n* = 27; 15.82 ± 8.19) to 3 months (*n* = 10; 19.82 ± 9.01) and 10.5 min (+66%) from 8 weeks to 6 months (*n* = 5; 26.31 ± 9.23). Overall, adherence was 82%. Patient satisfaction using the 5-point Likert scale scored 4.27 (465 answers in total). No adverse events were reported by participants in the tele-rehabilitation with VAPA group during the trial (Appendix A).

### 3.6. Additional Results

Results regarding exclusion criteria (Appendix A); reasons for dropouts (Appendix A); participants vs. non-participants (Appendix A); baseline data (Appendix A); and follow-up data (Appendix A); confidence interval of other variables (Appendix A); non inferiority 8 weeks vs 3- and 6 months follow-up (Appendix A); and exercise time, adherence and patient satisfaction in the telerehabilitation with VAPA group (Appendix A).

## 4. Discussion

The current study assessed VAPA, a new digital platform for tele-rehabilitation in COPD patients as an alternative to hospital-based rehabilitation. VAPA tele-rehabilitation was found to be non-inferior to traditional hospital rehabilitation in terms of sustained exercise capacity, and patient exercise tolerance, adherence, and satisfaction, and there were no safety issues. With respect to exercise capacity, tele-rehabilitation was non-inferior compared to standard rehabilitation based on the minimal important clinical difference of 35 m in the 6MWT endpoints [43]. During follow-up, however, we noticed a numerical trend toward improved performance with tele-rehabilitation with VAPA compared to traditional rehabilitation, though it was not statistically significant. Patients who were randomly assigned to the VAPA tele-rehabilitation program experienced an increase in training time at the follow-up visits, as also observed by Vogiatzis et al. [46].

Previous tele-rehabilitation studies in COPD [15,16] have failed to show the superiority of tele-rehabilitation compared to rehabilitation using a videoconference-based platform and group tele-training. In these studies, it was expected that patients allocated to the intervention group would train more often compared to those in the standard rehabilitation group, which could not be demonstrated. Therefore, our study was designed to show non-inferiority between VAPA tele-rehabilitation and standard rehabilitation.

We found no statistical difference in 7-day pedometry or VMCPM between groups or over time in each group, although we saw a trend towards better performance with standard rehabilitation. This may in part be inherent in that patients in the control group had to visit the hospital twice a week during the first part of the trial. However, their pedometry data decreased at a similar pace compared to VAPA tele-rehabilitation. The large standard deviation in both groups may be explained in part by the number of patients and the fact that patients began the research at different seasons of the year, since patients entered during the summer may have walked more than those enrolled during the winter.

We tried to digitize as much of the rehabilitation program’s information as feasible at VAPA and combined exercise training with empowering e-learning packages on how to live with COPD. There was no statistically significant difference between the groups in terms of QoL, i.e., the impact of digital e-learning and training was non-inferior to standard rehabilitation. These results suggest that the approach executed with e-learning was successful.

There are a variety of technologies and tele-rehabilitation platforms to choose from. The majority rely on videoconferencing, with one or more patients practicing in real time with a therapist, or virtual reality technologies, with agents demonstrating the exercises. [17,25]. Patients can stay at home and meet electronically using video platforms such as those used in prior trials [15,16,17,18,19].

The aim of this technique is to allow patients to engage in pulmonary rehabilitation in the hospital rehabilitation unit over the internet. The second aim is to maintain telemediated training as a social event similar to meeting face-to-face at the hospital. The intention is to support the social needs of patients training in a virtual group similar to group training in standard rehabilitation programs. Patients benefit from this setup since they can train at home, but training is still planned at specific times, exactly like in the hospital setting, making it less flexible than the VAPA platform. In contrast to VAPA, where patients can train whenever they choose, patients must still schedule their day and energies around the training appointment [18]. It is unclear whether a lack of socialization with other patients has an impact on QoL. Patients who trained with a “agent tele-rehabilitation set-up” in our study continued to train with VAPA more frequently and for longer periods of time in the follow-up phase, thus potentially maximizing the benefits of the treatment [47]. In our study, patient adherence was moderate to high (82%) and far more than the 20–50% found in earlier investigations [47,48]. Patient satisfaction was rated as very good and similar to the findings of other investigations [14]. VAPA has the capacity to match the exercise intensity to the patient’s pulse in real time and guide the thresholds during each exercise session, and this may account for the greater beneficial behavioral improvements observed with VAPA. In accordance with previous studies [49], no adverse events were reported.

The patient drop-out rate in tele-rehabilitation with VAPA was lower than in the standard rehabilitation group but similar to the dropout rate that Sohanpal et al. found in their review on self-management support programs for COPD patients [3].

There are several strengths of our study, including the randomized design and the long-term follow-up. Our study also has several limitations. The power calculations based on 6MWT do not allow the calculation of non-inferiority for the secondary parameters. The high drop-out rate, which is potentially attributable to a lack of commitment and 9transportation issues in standard rehabilitation and the small number of randomized participants that may have caused participants receiving tele-rehabilitation with VAPA to be younger, may have impacted the results. Patients with severe and advanced disease, on the other hand, are more likely to drop out, and our drop-out percentage is comparable to that reported in rehabilitation programs for patients with other chronic respiratory disorders [50,51].

## 5. Conclusions

Tele-rehabilitation with VAPA is non-inferior with respect to exercise capacity in patients with COPD when compared to a standard 8-week rehabilitation program. Tele-rehabilitation patients had a relatively high exercise time, high adherence and high patient satisfaction, and participation was without risk. Tele-rehabilitation with VAPA is a viable alternative rehabilitation approach for COPD patients, as well as a potentially effective tool for increasing COPD positive behavioral change toward a more physically active lifestyle.

## Figures and Tables

**Figure 1 jcm-11-00011-f001:**
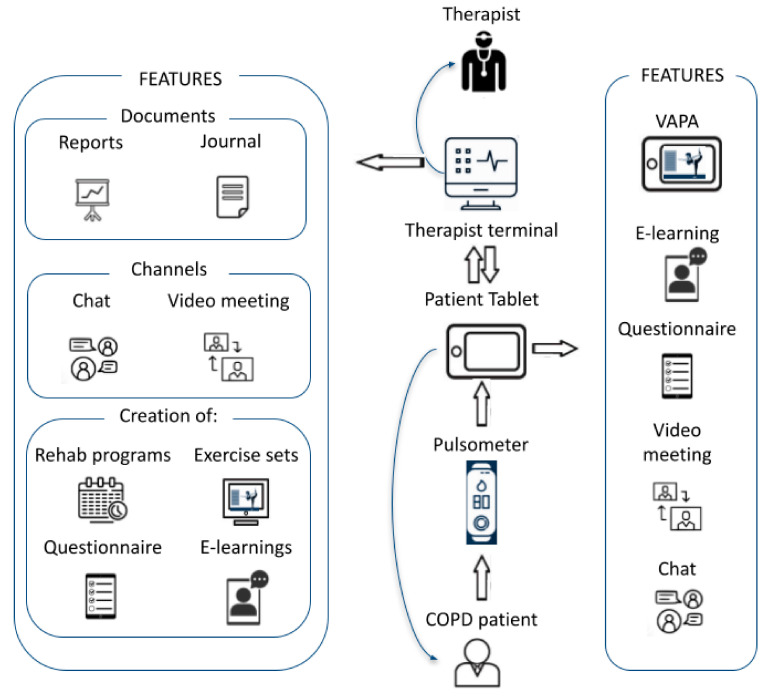
The multifaceted VAPA platform and its digital environment (from Cerdan et al. [27]) (Figure 1 is taken from the paper “Tele-Rehabilitation Program in Idiopathic Pulmonary Fibrosis—A Single-Center Randomized Trial by Cerdan-de-las-heras et al. [27], used under CC BY [28]/content modified from original).

**Figure 2 jcm-11-00011-f002:**
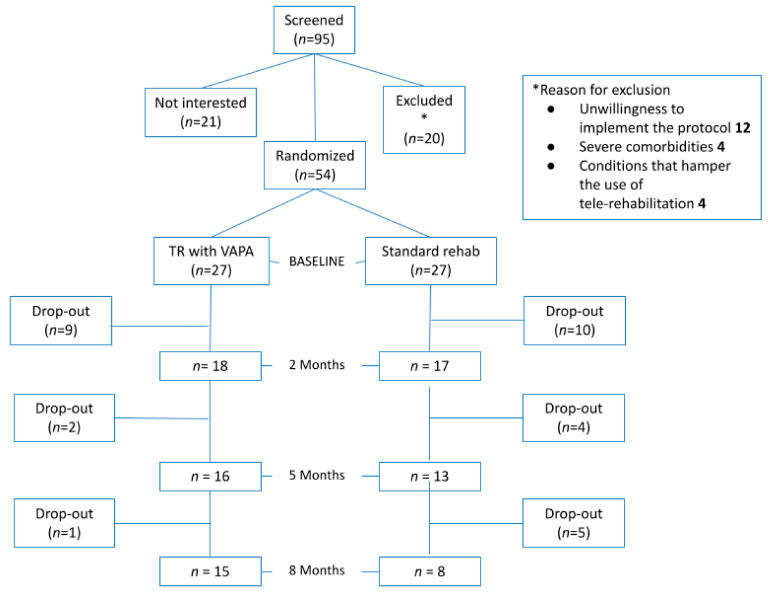
Randomization and enrollment in the general population.

**Figure 3 jcm-11-00011-f003:**
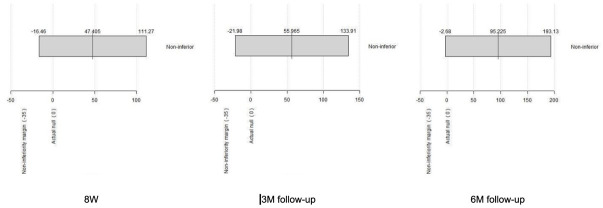
Non-inferiority test regarding the 6 min walk test distance between baseline and after 8 weeks of rehabilitation and after 3 and 6 months of follow-up. Data are shown as the mean difference between groups and 95% confidence intervals.

**Table 1 jcm-11-00011-t001:** The tele-rehabilitation program. (from Cerdan et al.). (Table 1 is taken from the paper “Tele-Rehabilitation Program in Idiopathic Pulmonary Fibrosis—A Single-Center Randomized Trial by Cerdan-de-las-heras et al. [27], used under CC BY [28]/content modified from original).

Features	Explanation
Workout Sessions with VAPA	The patients trained 10–20 min 3–5 times a week at home with their individual and tailored VAPA using training aids, such as elastics, weights and a fitness step, to reach the highest workout intensity. The VAPA provided encouragement to continue training during the workout based on a decision support system collecting real time data from a biometric sensor attached to the patient’s chest. The decision support system follows, in real time, heart rate data tracked by a biometric sensor attached to the chest of the patient, and according to different parameters, such as age, gender and medication, adjusts the training intensity with easy–difficult exercises used in hospital-based rehabilitation, adapted for home-base execution and stimulating the patient’s aerobic–anaerobic workout.
E-Learning Packages	The patient had access to e-learning packages addressing psychological, medical, nutritional and physical aspects of COPD—in part supplied by relevant special data sources medicin.dk [29], lunge.dk [30] and helbredsprofilen.dk. [31], or created by dietitian students after in-depth interviews with pulmonary patients [32,33,34,35].
Questionnaires	The patients filled out questionnaires regarding satisfaction, breathlessness, and adverse events reporting.
Video Consultation Sessions	Each patient met the physiotherapist in a video consultation to plan the rehabilitation program and to evaluate previous training experience.
Chat Sessions	Allowed the patient to interact with and obtain prompt answers from the physiotherapist.

**Table 2 jcm-11-00011-t002:** Demographics of the 54 patients that took part in the trial at baseline.

Parameters	TR with VAPA *n* = 27	Standard Rehab *n* = 27	*p*
Male, *n* (%)	16 (51.6)	15 (48.4)	-
Age (years) *	67.4 (10.2)	72.5 (7.4)	0.04
Smoking Status ^§^			
*Current, n (%)*	4 (16)	3 (14.3)	-
*Former, n (%)*	21 (84)	18 (85.7)	-
*Never, n (%)*	0 (0)	0 (0)	-
Long-Term Oxygen Therapy, *n* (%)	2 (7.4%)	2 (7.4%)	-
FVC (% predicted) *	67.4 (19.9)	70.2 (17.9)	0.60
FEV1 (% predicted) *	36.1 (14.1)	32.8 (8.5)	0.31
FEV1 Ratio (% predicted) *	48.6 (15.4)	39.1 (17.5)	0.04
6MWTD (m)*	385.5 (86.9)	366.6 (97.8)	0.46
7-Day Pedometry *	8601 (4831)	9234 (7126).	0.71
7dVMCPM *	282.1 (133)	358.3 (262)	0.19
SGRQ total *	55.6 (13.5)	60.6 (14.1)	0.03
SGRQ, Symptoms *	56.15 (21.1)	61.2 (23.5)	0.85
SGRQ, Activity *	77.5 (14.35)	76.8 (15.4)	0.05
SGRQ, Impact *	42.8 (15.5)	51.2 (15.9)	0.18
IADL *	1.1 (1.1)	2.2 (2.3)	0.46
GAD7 *	3.3 (3.9)	5.9 (6.6)	0.41

* Mean (SD); ^§^ missing smoking journal status of 8 patients; FVC: forced vital capacity; FEV1: forced expiratory volume in the first second; 6MWTD: distance walked during the 6 min walk test; 7dVMCPM: 7-day vector magnitude counts per minute; SGRQ: Saint George Respiratory Questionnaire; IADL SCORE: Instrumental Activities of Daily Living Scale; GAD7: General Anxiety Disorder-7 Questionnaire.

## Data Availability

Data are contained within the article or Appendix A.

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
