# Peer review of "Effect of a New Tele-Rehabilitation Program versus Standard Rehabilitation in Patients with Chronic Obstructive Pulmonary Disease"

_jcm, 2021, doi:10.3390/jcm11010011_

Round 1

Reviewer 1 Report

Dear Authors, 

Thank you for the opportunity to review the paper entitled “Effect of a New Tele-rehabilitation Program versus Standard Rehabilitation in Patients with Chronic Obstructive Pulmonary Disease”. The contents of the manuscript are very interesting and timely. The study was aimed at the effect of a tele-rehabilitation program vs. standard rehabilitation in COPD. The design of the paper and the methodology of the study are correct, however, they need a broader description. Below is a list of my comments:

Major:

  • I think the introduction is too short, does not describe the background of what telerehabilitation might bring, the types of telerehabilitation, and also does not address the changes in patient management during a pandemic - please expand.
  • How was the training intensity evaluated? Table 1 indicates that ... "to reach the highest workout intensity" what does this mean? Has the HR been evaluated?
  • No effect size was indicated for each outcome

Minor:

  • Wrong reference style
  • I think the 16 reference is unnecessary
  • Report 2 and 3 of the supplementary documents seem to be redundant.
  • I would suggest in Table 2, adding a column with the p values of the between-group analysis instead of â™± p≲05;
  • No effect size was indicated for each outcome
  • The abbreviation VMCPM is not explained
  • An explanation of the contents of the supplementary materials seems unnecessary (lines 277-318).

Author Response

Dear reviewer, thank you for your time and efforts helping us to improve this paper. Please find below a resume of the different changes we have executed in the paper according to your review:

  1. I think the introduction is too short, does not describe the background of what telerehabilitation might bring, the types of telerehabilitation, and also does not address the changes in patient management during a pandemic - please expand.
    1. Thank you again for this remark. In order to improve the paper and according to your comment, we have added the following text to the introduction. We hope you find it interesting and sufficient:

In a recent review, the usage of tele-rehabilitation in COPD demonstrates potential effectiveness, high patient acceptance, and strong motivation to engage patients in physical activity [1]. Tele-rehabilitation is a method that is used today to treat, test and follow patients from a distance in order to empower them to cope with their short- and long-term impairments and help them to be physical, mental, emotional, vocational and social independent, thus improving and maintaining their quality of life. In another review, five different methods of tele-rehabilitation in pulmonary diseases are described: 1-videoconferencing, 2-telephone, 3-using a website with telephone support, 4-using a mobile application and 5- text message support with a mobile app [2]. Tele-rehabilitation was born with a first aim to reduce the hospitalization time and to treat patients in rural areas [3] but now, in the COVID-19 pandemia times, it is introduced as an alternative to conventional rehabilitation and as an action to prevent infections and support continuity of rehabilitation [4–6] 

References:

1. Rutkowski, S. Management Challenges in Chronic Obstructive Pulmonary Disease in the COVID-19 Pandemic: Telehealth and Virtual Reality. J. Clin. Med. Res. 2021, 10, doi:10.3390/jcm10061261.

2. Cox, N.S.; Dal Corso, S.; Hansen, H.; McDonald, C.F.; Hill, C.J.; Zanaboni, P.; Alison, J.A.; O’Halloran, P.; Macdonald, H.; Holland, A.E. Telerehabilitation for Chronic Respiratory Disease. Cochrane Database Syst. Rev. 2021, 1, CD013040.

3. Peretti, A.; Amenta, F.; Tayebati, S.K.; Nittari, G.; Mahdi, S.S. Telerehabilitation: Review of the State-of-the-Art and Areas of Application. JMIR Rehabil Assist Technol 2017, 4, e7.

4. Why Is Telerehabilitation so Important during the COVID-19 Pandemic? Available online: https://shrs.uq.edu.au/article/2020/05/why-telerehabilitation-so-important-during-covid-19-pandemic (accessed on 3 November 2021).

5. Scherrenberg, M.; Frederix, I.; De Sutter, J.; Dendale, P. Use of Cardiac Telerehabilitation during COVID-19 Pandemic in Belgium. Acta Cardiol. 2021, 76, 773–776.

6. Scherrenberg, M.; Wilhelm, M.; Hansen, D.; Völler, H.; Cornelissen, V.; Frederix, I.; Kemps, H.; Dendale, P. The Future Is Now: A Call for Action for Cardiac Telerehabilitation in the COVID-19 Pandemic from the Secondary Prevention and Rehabilitation Section of the European Association of Preventive Cardiology. European Journal of Preventive Cardiology 2021, 28, 524–540.

  1. How was the training intensity evaluated? Table 1 indicates that ... "to reach the highest workout intensity" what does this mean? Has the HR been evaluated?
    1. Thank you again for this question. This training intensity is evaluated and calibrated via a decision support system (DSS) designed by Physio R&D, the company behind the tele-rehabilitation platform used in this study. We are not able to explain this in more details except that DSS follows, in real time, heart rate data tracked by a biometric sensor attached to the chest of the patients, and according to different parameters as age, gender and medication adjust the training intensity with easy - difficult exercises used in hospital-based rehabilitation, adapted for home-base execution and stimulating patient’s aerobic-anaerobic workout.  For that we have added this text to the table 1:

...The decision support system follows, in real time, heart rate data tracked by a biometric sensor attached to the chest of the patients, and according to different parameters as age, gender and medication adjust the training intensity with easy - difficult exercises used in hospital-based rehabilitation, adapted for home-base execution and stimulating patient’s aerobic-anaerobic workout.

  1. No effect size was indicated for each outcome
    1. Thank you again for this remark. As we understand it is just the first outcome, the one that needs to have a certain effect size. This is specified in  2.6. Statistics as “...minimally important clinical difference of 35m in the 6MWT distance....”. We do not see any need in specifying effect size for the rest of the outcomes when the power calculation for this study has been based solely on non-inferiority for exercise capacity based on the 6MWT but are happy to learn more. For us the rest of the outcomes have been helpful to understand how those variables behave during tele-rehabilitation with VAPA. We do not find a need to add extra text to the paper about this and we hope you will agree on that.
  2. Wrong reference style
    1. Thank you for making us aware of this, We have changed the reference style.

  1. I think the 16 reference is unnecessary You mean this one: 

16. Guidelines for Pulmonary Rehabilitation Programs Available online: https://books.google.com/books/about/Guidelines_for_Pulmonary_Rehabilitation.html?id=kuqODwAAQBAJ (accessed on 10 September 2020).

  1. Thank you again for this remark, we discovered that the link is not working anymore and we changed the reference to this new reference:

22. Bolton, C.E.; Bevan-Smith, E.F.; Blakey, J.D.; Crowe, P.; Elkin, S.L.; Garrod, R.; Greening, N.J.; Heslop, K.; Hull, J.H.; Man, W.D.-C.; et al. British Thoracic Society Guideline on Pulmonary Rehabilitation in Adults: Accredited by NICE. Thorax 2013, 68, ii1–ii30.

  1. Report 2 and 3 of the supplementary documents seem to be redundant.
    1. Thank you again for this remark. Report 1 compares baseline data for patients who were found eligible for trial participation and declined to participate with the patients that accepted to participate. However, we have followed your advice and deleted both.

Report 2 and 3 are deleted from the Supplementary.

  1. I would suggest in Table 2, adding a column with the p values of the between-group analysis instead of ♱ p≲05;
    1. Thank you again for this remark. We added the p values of the relevant variables.
  2. No effect size was indicated for each outcome
    1. Thank you again for this remark. Please see the answer above.
  3. The abbreviation VMCPM is not explained
    1. Thank you again for this remark. ​​VMCPM means “vector magnitude counts per minute” and it is an activity score that follows steps walked but also measures the intensity of the movements using the same pedomtry sensor. This abbreviation has been added in 2.5. Endpoints.page 
  4. An explanation of the contents of the supplementary materials seems unnecessary (lines 277-318).
    1. Thank you again for this remark. We understand you are talking about this text in the article: 

“...Supplementary Materials[M1] : The following are available online at (www.mdpi.com/xxx/s1---) 277 Report 1 (exclusion criteria); Report 2 (Participants non participants): Table 1: Analysis of differences 278 between non-participants and participants; Report 3 (Baseline data): Table 1: Baseline de-279 mographics, Table 2: Baseline demographics in the control and intervention group ; Report 4 (Fol-280 low-up data) :Table 1. Data at follow-up after 8 weeks of training. Analyses are shown with an in-281 dependent t-test within groups, Figure 1. Mean difference and 95% CI for 6MWT between patients 282 in the control and TR with VAPA groups, Table 2. Data at follow-up three months after completion 283 of training. Analyses are shown with an independent t-test within groups, Figure 2. Mean difference 284 and 95% CI for 6MWT between patients in the control and TR with VAPA groups, Table 3. Data at 285 follow-up six months after completion of training. Analyses are shown with an independent t-test 286 within groups, Figure 3. Mean difference and 95% CI for 6MWT between patients in the control and 287 TR with VAPA groups, Figure 4 – Mean and Standard Deviation for the 6 minutes walking test over 288 time for patients in the control and intervention group (meters), Figure 5. Mean and Standard Devi-289 ation for the 7 days pedometer over time for patients in the control and intervention group (meters), 290 Figure 6. Mean and Standard Deviation for the 7 dVMCPM over time for patients in the control and 291 intervention group (meters), Figure 7. Mean and Standard Deviation for the pulmonary function 292 FVC% over time for patients in the control and intervention group (percentage), Figure 8. Mean and 293 Standard Deviation for the pulmonary function FVE1% over time for patients in the control and 294 intervention group (percentage), Figure 9. Mean and Standard Deviation for the pulmonary func-295 tion FVE1/FVC ratio over time for patients in the control and intervention group (percentage), Fig-296 ure 10. Mean and Standard Deviation for the SGQR over time for patients in the control and inter-297 vention group, Figure 11. Mean and Standard Deviation for GAD-7 over time for patients in the 298 control and intervention group, Figure 12. Mean and Standard Deviation for the Instrumental Ac-299 tivities Of Daily Living Scale over time for patients in the control and intervention group, Figure 13. 300 Mean and Standard Deviation for the 4 meters gait test over time for patients in the control and 301 intervention group, Table 4: Analysis of changes over time in pulmonary function, physical perfor-302 mance, physical activity, exercise recovery and quality of life in the control group. (P-value for base-303 line compared with 8 weeks, baseline compared with 3 months and baseline compared with 6 304 months), Table 5: Analysis of changes over time in pulmonary function, physical performance, phys-305 ical activity, exercise recovery and quality of life in the intervention group. (P-value for baseline 306 compared with 3 months, baseline compared with 6 months and baseline compared with 9 months). 307 Report 5 (Confidence interval of other variables): Table 1: Quality of life differences between groups 308 in each evaluation, mean with 95% confidence intervals and p-value between baseline vs 8 weeks; 309 3-, and 6 months follow-up, Table 2: Seven days pedometry and vector magnitude counts per mi-310 nute differences between groups at each follow-up, mean with 95% confidence intervals and p value 311 between baseline vs 8 weeks; 3-; and 6 months follow-up; Report 6 (Non inferiority 8 weeks vs 3- 312 and 6 months follow-up): Table 1. Three months follow-up difference after treatment, Table 2. Six 313 months follow-up difference after treatment; Report 7 (Exercise Time, Adherence and Patient Satis-314 faction in the Telerehabilitation with VAPA group): Figure 1: Exercise time by patients in the TR 315 with VAPA. Data is shown as the average time (min) per exercise session (excluding pause time) 316 and Table 1: Patient adherence and satisfaction shown as the training time expected vs performed 317 and patient satisfaction from baseline to follow-up after 8 months.  ..”

We think the same as you, but this is the rule in this journal so we keep it as they expected. But if the editor accepts to resume it, then we would like to change it on this way:

Supplementary Materials[M1] : The following are available online at (www.mdpi.com/xxx/s1---) Report 1 (Exclusion criteria); Report 2 (Follow-up data); Report 3 (Confidence interval of other variables); Report 4 (Non inferiority 8 weeks vs 3- and 6 months follow-up); Report 5 (Exercise Time, Adherence and Patient Satisfaction in the Telerehabilitation with VAPA group); Report 6 (Reasons for dropout)

Please do not hesitate to contact us again if our actions to your comments are not as expected.

Thank you in advance. 

Jose Cerdan, on behalf of the authors of the article.

Reviewer 2 Report

The authors report data of a randomized non-inferiority study comparing tele-rehabilitation with standard rehabilitation in COPD. Despite the appealing methodology, the current manuscript has important limitations:

  1. The title is misleading. In fact the authors describe 2 forms of reactivation in COPD patients. Indeed, rehabilitation starts by definition by a thorough assessment of the experienced limitations by the patient. Only generic information is provided about activity parameters. No information is provided about limiting causes of exercise intolerance in these severe COPD population. No information is provided about training intensity in both programs. It can be questioned whether the standard program can result in any training effect, considering frequency and duration of training.

2. The authors over interpret the data 3 and 6 months based on the power calculation and the number of drop outs.

3. Design. It seems that the study is rather poorly designed as no maintenance program is provided in the standard rehab group. Despite the low number of subjects, randomization resulted in important differences between both groups. No information is provided about blindness of the outcome measurements. No information is provided about reasons for drop out.

4. Patient characteristics. Besides the parameters assessed in table 2, it would be very helpful when other parameters are included: degree of hyperinflation, degree of gas exchange impairment, cardiovascular evaluation????

5. Drop out interpretation: in fact, drop out in the first 8 weeks is similar between both groups. Only after this period, drop out is higher after standard rehabilitation.

6. Results: continued use of VAPA  can not be considered as an outcome of this intervention.

7. Table 1: e-learning. It seems that the authors have described the wrong modules? IPF= interstitial pulmonary fibrosis?? Are these e-learing packages scientifically validated?largely based on contribution by students?

Author Response

Dear reviewer, 

Thank you for your time and efforts in helping us to improve our paper. Please find below a resume of the different changes we have added to the paper according to your review:

  1. The title is misleading. In fact the authors describe 2 forms of reactivation in COPD patients. Indeed, rehabilitation starts by definition by a thorough assessment of the experienced limitations by the patient. Only generic information is provided about activity parameters. No information is provided about limiting causes of exercise intolerance in these severe COPD population. No information is provided about training intensity in both programs. It can be questioned whether the standard program can result in any training effect, considering frequency and duration of training.

Thank you for your feedback with this remark. You are right that “...rehabilitation starts by definition by a thorough assessment of the experienced limitations by the patient…” but this is something outside this trial. All the assessments you point to were performed before the patients were referred to the rehabilitation program in the hospital and before they were recruited for this trial. 

Regarding your comment “...Only generic information is provided about activity parameters…”  yes, this is true, and we used the information from the patient electronic file collected in the thorough assessment relevant for this study, “The 6 minute walk test distance” as is also our primary outcome in this study. No other activity data was collected due to the fact no other activity tests were performed in these patients according to the treatment protocol at Aarhus University Hospital. We did not collect extra activity data at baseline but used the data collected as specified in the hospital protocol for COPD patients before inviting them to a rehabilitation program.

Regarding your comment “...No information is provided about limiting causes of exercise intolerance in these severe COPD population …”, thank you for this remark. We think this is outside of the scope of this study. Our aim is to compare tele-rehabilitation with normal treatment and not to investigate the limiting causes of exercise intolerance. We think it is an interesting topic, but not for the scope of this study and have not elaborated on this topic. We hope you agree.

Regarding your comment “... No information is provided about training intensity in both programs…” We thank you for your remark. Regarding the intensity for the control group, we previously referred to the standard protocol of rehabilitation using this reference:

  1. Guidelines for Pulmonary Rehabilitation Programs Available online: https://books.google.com/books/about/Guidelines_for_Pulmonary_Rehabilitation.html?id=kuqODwAAQBAJ (accessed on 10 September 2020).” However, this linking does no exist and has been changed to this new reference:

22. Bolton, C.E.; Bevan-Smith, E.F.; Blakey, J.D.; Crowe, P.; Elkin, S.L.; Garrod, R.; Greening, N.J.; Heslop, K.; Hull, J.H.; Man, W.D.-C.; et al. British Thoracic Society Guideline on Pulmonary Rehabilitation in Adults: Accredited by NICE. Thorax 2013, 68, ii1–ii30.

Regarding the intensity for the intervention group, we are grateful for your comments and have added this text to Table 1:

...The decision support system follows, in real time, heart rate data tracked by a biometric sensor attached to the chest of the patients, and according to different parameters as age, gender and medication adjust the training intensity with easy to difficult exercises used in hospital-based rehabilitation, adapted for home-base execution and stimulating patient’s aerobic-anaerobic workout.

Regarding your comment “...It can be questioned whether the standard program can result in any training effect, considering frequency and duration of training …”

We think this comment is answered by referring to the rehabilitation protocol above

  1. The authors over interpret the data 3 and 6 months based on the power calculation and the number of dropouts.
    1. Thank you again for this remark. We do not agree with you in the sense that we just focus on a non inferiority test comparing the baseline with 8 weeks of tele-rehabilitation, 3 and 6 months follow up and as long as the patient group had a minimally important clinical difference of 35m in the 6MWT distance compared to baseline, for us this mean the telerehabilitation with VAPA was non inferior to normal treatment. 
  2. Design. It seems that the study is rather poorly designed as no maintenance program is provided in the standard rehab group. Despite the low number of subjects, randomization resulted in important differences between both groups. No information is provided about blindness of the outcome measurements. No information is provided about reasons for drop out.
    1. Thank you again for this remark. Yes, we agree, with regards to ”no maintenance program is provided in the standard rehab group”. Maintenance training is not part of the standard rehabilitation program in the hospital but patients are encouraged to continue training. Both groups were encouraged to continue training beyond the 8 weeks program. However, we found it interesting to follow those in the intervention group that continued using VAPA telerehabilitation for a longer period without the intervention of a real therapist opposed to the first 8 weeks. Any participant in both groups could potentially continue training 1) by themselves, 2) continue in a municipality- or local community- organized training, 3) stop training or 4) in the case of the intervention patients, continue training with VAPA. We had no opportunity to follow or track the training behavior of the participants after finishing the 8 weeks rehabilitation besides the follow-up visits planned.  
    2. Regarding your comment “...Despite the low number of subjects, randomization resulted in important differences between both groups …” yes, it is true and a pity it happened. We have expressed our concerns in the discussion“...The high drop-out rate, which is potentially attributable to a lack of commitment and transportation issues in standard rehabilitation and the small number of randomized participants that may have caused participants receiving tele-rehabilitation with VAPA to be younger, may have impacted the results …” 
    3.  Regarding your comment “...No information is provided about blindness of the outcome measurements …” We added the text below to the article in the 2. Methods and Materials 2.1. Design of the Research.

“... The different objective tests were executed by an independent research nurse not involved in the study…”

  1. Regarding your comment “... No information is provided about reasons for drop out …” Thank you again for your comment. We have added this information to the supplementary as report 6 and refferend in the article in 3. Results 3.1. Patients

Article extra text:

“... Reasons for dropout are expressed in report 6 in supplementary. …”

Supplementary:

Report 6 (Reasons for dropout)

  • Personal causes 7
  • Worsening condition 5
  • Not following training protocol 4
  • Not attended 8M follow up 4
  • Not attended 5M follow up 2
  • Transportation problems 2
  • No surplus to continue 2
  • Can not see improvements 1
  • Had an accident 1
  • Sensor problems 1
  • Misunderstanding study goal 1
  • Dead 1

  1. Patient characteristics. Besides the parameters assessed in table 2, it would be very helpful when other parameters are included: degree of hyperinflation, degree of gas exchange impairment, cardiovascular evaluation????
    1. Thank you again for this remark. These parameters are indeed interesting to study but are not part of the usual investigational program for COPD patients at the hospital unless lung volume reduction or lung transplantation is considered. Thus, we do not have these data but will consider to include them in future studies 
  2. Drop-out interpretation: in fact, drop out in the first 8 weeks is similar between both groups. Only after this period, drop out is higher after standard rehabilitation.
    1. Thank you again for this remark. We discussed the dropout in the discussion with this text “...Patients with severe and advanced disease, on the other hand, are more likely to drop out, and our drop-out percentage is comparable to that reported in rehabilitation programs for patients with other chronic respiratory disorders [51,52] …” We do not find that we claim something wrong compared to your remark. The point is that severe COPD patients usually have a high drop-out rate in many other studies similar to what we have seen in this study. We hope the text in the discussio is sufficient.
  3. 6. Results: continued use of VAPA cannot be considered as an outcome of this intervention.
    1. Thank you again for this remark. We agree that continued use of VAPA was not a pre-specified outcome, but we still find it interesting to report, also because patients asked to continue the training program even without a real physiotherapist after the first 8 weeks. We hope that you agree with us.
  4. Table 1: e-learning. It seems that the authors have described the wrong modules? IPF= interstitial pulmonary fibrosis?? Are these e-learning packages scientifically validated? largely based on contribution by students?
    1. Thank you again for this remark. We made a mistake in writing IPF and not COPD in the text of Table 1. The e-learning packages were in fact developed for patients with COPD. The packages were created based on high quality videos developed by well renowned organisations in healthcare or students that had in-depth interviews to understand better what was important for this patient group to know and how this information should be given and understanding measured.  For us this part of the project was explorative, as we tried to have similar actions as in the normal rehabilitation treatment, where patients usually participate in 8 one-hour workshops in their rehabilitation program (at least in Århus University Hospital) with different specialists to better learn and understand their disease and get instructions on how to cope with symptoms in different situations of their daily life. We decided to convert the face 2 face workshops with e-learning for the tele-rehabilitation group. The process was,1-interesting videos were collected by the pulmonary department nurses from different well renowned organisations, 2- if any topic of the workshops was not found in qualitative videos, master students were asked to create them. Videos collected or created were screened by head department doctors and from there e-learning packages were drafted and discussed internally in the pulmonary department before releasing them officially in the tele-rehabilitation program.  We have decided to not add all this information to the paper as we want to keep the red line to the first outcome, so because of that we simplified the information written in table 1. We hope you agree with this decision. No further changes has been added to the text 

Please do not hesitate to contact us again if our actions to your comments are not as expected.

Thank you in advance. 

Jose Cerdan, on behalf of the authors of the article.

Round 2

Reviewer 2 Report

The reviewer thanks the authors for the reply. Some comments are adequately addressed but the more principle issues remain unaddressed: formulation that the comments are outside the scope of the paper seems a matter to avoid the content of the comments.

Detailed: 

1. Based on the comments of the authors, it is clear for me that they focus on the application of telemedicine as a tool to reactivate in this case COPD patients.  When the authors are willing to change the title accordingly and make this notion in the manuscript, I will have no further comments.

2. However, the authors formulate the current intervention as a form of “pulmonary rehabilitation”. For that reason, I addressed in my comments why the intervention in my opinion not fulfills the criteria of “pulmonary rehabilitation” and that even the outcomes in the standard intervention can be questioned. Asking for specific traits determining exercise limitation, the authors reply that assessments are performed “before” referral to the intervention program. Without such information, it will be difficult to interpret the conclusions of the manuscript.

For all these reasons, my conclusion is that the scope and conclusions of the manuscript must be downsized to the feasibility of tele-intervention to reactivate COPD patients.

I fully agree that the authors will reply that the delineation of pulmonary rehabilitation is not very sharp. 

Author Response

Dear Editor,

Our response to 2nd reviewer is in the attached word file.

We hope for 2nd reviewer's understanding and that she/he can accept our explanations and further publication of the paper.

Thank you in advance

The co-authors team.
